# Glucuronides Hydrolysis by Intestinal Microbial *β*-Glucuronidases (GUS) Is Affected by Sampling, Enzyme Preparation, Buffer pH, and Species

**DOI:** 10.3390/pharmaceutics13071043

**Published:** 2021-07-08

**Authors:** Christabel Ebuzoeme, Imoh Etim, Autumn Ikimi, Jamie Song, Ting Du, Ming Hu, Dong Liang, Song Gao

**Affiliations:** 1Department of Pharmaceutical Science, College of Pharmacy and Health Sciences, Texas Southern University, 3100 Cleburne Street, Houston, TX 77004, USA; c.ebuzoeme9304@student.tsu.edu (C.E.); i.etim9944@student.tsu.edu (I.E.); a.ikimi2305@student.tsu.edu (A.I.); du.ting@tsu.edu (T.D.); dong.liang@tsu.edu (D.L.); 2College of Liberal Arts, The University of Texas at Austin, Austin, TX 78712, USA; js83847@utexas.edu; 3Department of Pharmacological and Pharmaceutical Sciences, College of Pharmacy, The University of Houston, 4901 Calhoun Street, Houston, TX 77204, USA; mhu@uh.edu

**Keywords:** microflora, glucuronides, hydrolysis, microbial *β*-Glucuronidases (GUS)

## Abstract

Glucuronides hydrolysis by intestinal microbial *β*-Glucuronidases (GUS) is an important procedure for many endogenous and exogenous compounds. The purpose of this study is to determine the impact of experimental conditions on glucuronide hydrolysis by intestinal microbial GUS. Standard probe 4-Nitrophenyl *β*-D-glucopyranoside (pNPG) and a natural glucuronide wogonoside were used as the model compounds. Feces collection time, buffer conditions, interindividual, and species variations were evaluated by incubating the substrates with enzymes. The relative reaction activity of pNPG, reaction rates, and reaction kinetics for wogonoside were calculated. Fresh feces showed the highest hydrolysis activities. Sonication increased total protein yield during enzyme preparation. The pH of the reaction system increased the activity in 0.69–1.32-fold, 2.9–12.9-fold, and 0.28–1.56-fold for mouse, rat, and human at three different concentrations of wogonoside, respectively. The Vmax for wogonoside hydrolysis was 2.37 ± 0.06, 4.48 ± 0.11, and 5.17 ± 0.16 μmol/min/mg and Km was 6.51 ± 0.71, 3.04 ± 0.34, and 0.34 ± 0.047 μM for mouse, rat, and human, respectively. The inter-individual difference was significant (4–6-fold) using inbred rats as the model animal. Fresh feces should be used to avoid activity loss and sonication should be utilized in enzyme preparation to increase hydrolysis activity. The buffer pH should be appropriate according to the species. Inter-individual and species variations were significant.

## 1. Introduction

The role of gut microflora in health has gained increasing attention in the past decades [1,2,3]. One of the major benefits from intestinal microflora is that the commensal bacteria could generate beneficial metabolites, such as cancer-preventive compound equol for daidzein and compound k from ginsenosides, for the host from dietary components through different biotransformation pathways [4,5,6]. These metabolites either accumulate in the gastrointestinal system or reach distant organs, which are associated with certain physiological or pathological effects (e.g., gastrointestinal inflammation and carcinogenesis [7]). Glucuronide hydrolysis is one of the major metabolic pathways catalyzed by microflora. Many nutritional dietary components exist as glycoside or glucuronide forms, which must be hydrolyzed before absorption. For example, it is well-known that dietary components existing as glucuronide or glycoside forms, such as soy isoflavone genistin, daidzin, are usually hydrolyzed into aglycones by intestinal microflora to facilitate absorption [8,9]. In addition, some important endogenous compounds (e.g., bilirubin) also undergo glucuronide hydrolysis pathways to form entero-hepatic recycling [10,11], which is an important physiological phenomenon. Glucuronide hydrolysis has shown to be promising as pharmaceutical scientists explore methods to design glucuronide pro-drugs for colon drug delivery and also for improving its water solubility since glucuronide hydrolysis by intestinal microflora is the key step towards the release of pro-drugs [12,13].

Glucuronide hydrolysis is widely known for its catalyzation by GUS, which exhibits in the intestinal microbiota; it is expressed in at least 279 GUS isoforms displayed in the gut microflora. The human microbiome GUS structures were reviewed, and the atlas was published recently [14,15,16]. However, determination of GUS activity using an in vivo model is complex because the hydrolysis product (i.e., aglycone) can be conjugated back into glucuronides by highly expressed UGTs in the GI tract. The enzyme-mediated reaction is a well-accepted in vitro model to evaluate enzyme activity. However, it is complex to isolate or purify GUS from microflora. Recombinant microbial GUS might be available, but a single GUS isoform could not reflect the reaction occurring in the gut as there are many isoforms. Theoretically, bacteria culture seems to be the best approach to study the total activity of GUS in microflora and it is feasible to obtain gut bacteria from feces. However, most gut bacteria can only be cultured under anaerobic conditions and some bacteria are not culturable even under anaerobic conditions [17], which limits the application. Thus, total proteins prepared from feces (e.g., fecal S9 fractions and fecal suspensions) are usually used as the enzyme sources within in vitro studies reported in literature to determine glucuronide hydrolysis in the gut [18,19,20,21]. Nevertheless, the conditions used in these studies were not standardized and the results from different research groups may not be comparable because different reaction conditions were used. Variation of experimental conditions include: (1) What is the optimum feces collection time? (2) What is the best enzyme preparation method? (3) What are the optimum conditions of the reaction buffers? (4) What is the variation across individual animals? (5) What is the variation across mice, rats, and humans?

The objective of this study was to determine the variation and to optimize the incubation conditions used within in vitro microflora-mediated glucuronide hydrolysis studies. Feces collection, buffer pH, individual variation, and species difference will be evaluated. We used 4-Nitrophenyl *β*-d-glucopyranoside (pNPG), a standard GUS activity probe [22,23], to determine the enzyme activity. In addition, we used wogonoside (Figure 1), which is a naturally occurring flavonoid glucuronide for which aglycone possesses different pharmacological efficacy (e.g., anti-inflammatory, anti-cancer, antiviral, and neuroprotective actions) [24,25], as a model compound to determine its hydrolysis potential in the gut.

## 2. Materials and Methods

### 2.1. Chemicals

Wogonoside, wogonin, and 4-Nitrophenyl *β*-d-glucopyranoside (pNPG) (purity > 98) were purchased from Sigma Aldrich (St. Louis, MO, USA). Monopotassium phosphate (KH2PO4), dipotassium phosphate (K2HPO4), and sodium chloride (NaCl) were purchased from VWR International (Radnor, PA, USA). KPI buffer was made using KH2PO4 and K2HPO4 and the pH was adjusted using sodium hydroxide and hydrochloride. Other chemicals (typically analytical grade or better) were used as received. C57BL6 mice (male, 8 weeks) were obtained from Jackson lab (Bar Harbor, ME, USA) and F344 rats (male, 8 weeks) were bought from Charles River (Wilmington, MA, USA).

### 2.2. Feces Collection

All animals were housed according to the provisions of the Animal Welfare Act, PHS Animal Welfare Policy, NIH Guide for the Care and Use of Laboratory Animals, and the policies and procedures of Texas Southern University (TSU) Institutional Animal Use and Care Committee (IACUC). All protocols were approved by the IACUC at TSU.

Mice and rats’ feces were collected for enzyme preparation. Briefly, the animals were fed a regular AIN-93 diet for at least one week after being received from the vendors. In order to collect feces, the cages were cleaned out, and feces were collected within 24 h unless we required feces older than 24 h. Human feces were collected from healthy female adult volunteers. The collected feces were prepared immediately.

Different time points were collected to determine the impact of collection time on GUS activity. Briefly, the animals were sacrificed to collect the fresh feces from the colon for fresh enzyme preparation. In order to collect day 1 feces, the rats and mice were housed in the cages with new bedding and feces were collected 24 h after changing the bedding for enzyme preparation. In order to collect day 7 feces, the animals were housed in the cages with new beddings for 24 h and then the animals were moved out and the feces were collected on day 7 for enzyme preparation.

### 2.3. Fecal Enzyme Preparation

The fecal enzyme was prepared using two different procedures. Briefly, the pooled feces (1 g) were mixed with 10 mL ice cold 50 mM KPI buffer solution (pH 7.4) and swirled vigorously for 60 s followed by centrifugation at 1000× *g* and 4 °C for 10 min for the sonication method. The pellets were further washed using 5 mL of 50 mM KPI buffer solution. The supernatant was homogenized and sonicated (550 W) in the ice water bath for 30 min and centrifuged at 9000× *g* at 4 °C for 15 min according to the reported procedures [18]. The final supernatant, which is called S9 fraction, was aliquoted and stored at −80 °C. Protein concentrations were determined by the BCA protein assay kit using bovine serum albumin as the standard.

For the suspension method (11), 1 g of feces was mixed with 10 mL of cold 50 mM KPI buffer solution (pH 7.4). The mixture was vigorously swirled for 60 s, homogenized, and centrifuged at 9000× *g* at 4 °C for 30 min. The supernatant was aliquoted and stored at −80 °C. Protein concentrations were determined by the BCA protein assay kit using bovine serum albumin as the standard.

### 2.4. pNPG Hydrolysis

The pNPG hydrolysis was evaluated following the procedure reported previously with slight modification [26,27,28]. Briefly, a reaction mixture (200 μL) in KPI buffer containing 50 μL of protein (1 mg/mL, final concentration 0.25 mg/mL) and 20 μL of pNPG (10 mM, final concentration 1 mM) in a 96-well plate was incubated in an incubator shaker at 37 °C for 30 min. Then, the reaction was terminated by adding 20 μL of 6% formic acid in acetonitrile and each experiment was conducted in triplicate. The plate was read using a microplate reader at 405 nm. The relative hydrolysis rate was evaluated using the OD values.

### 2.5. Wogonoside Hydrolysis

The incubation procedures for measuring hydrolysis enzyme’s activities followed a similar procedure used in the previous publications [18]. Briefly, certain volume of the thawed fecal enzymes was transferred into 50 mM KPI buffer to achieve a final concentration of 10 μg/mL. Wogonoside was added to the system to achieve certain concentrations. The system (100 μL) was then incubated at 37 °C for 30 min in a water bath with an orbital shaker. The reaction was stopped by adding 25 μL of 6% formic acid in acetonitrile and followed by centrifugation (14,000× *g*, 15 min, 4 °C). The supernatant (10 μL) was injected into UPLC for analysis.

### 2.6. Quantification of Wogonoside and Wogonin Using UPLC

Wogonin and wogonoside were analyzed by a common UPLC chromatographic method: system, Waters Acquity UPLC with photodiode array detector and Empower software; column, BEH C18, 1.7 μm, 2.1 × 50 mm; mobile phase B, 100% acetonitrile and mobile phase A, 0.1% formic acid in water (pH 2.5); flow rate of 0.5 mL/min; gradient, 0 to 1.0 min, 10–50% B, 1 to 1.5 min, 50–55% B, 1.5 to 2.5 min, 55–70% B, 2.5 to 2.8 min, 70–10%, and 2.8 to 3.0 min, 10–10% B; detection wavelength, 273 nm; and injection volume, 10 μL. Standard curve samples containing wogonoside and the metabolite wogonin in the same matrix were prepared and injected into UPLC for quantification. Rutin (0.2 μM in acetonitrile) was used as the internal standard.

### 2.7. Optimization of Incubation Buffer and MgCl_2_ Concentration

In order to determine the impact of Mg^2+^ on GUS activity, pNPG was incubated with fecal enzymes in KPI buffer containing 0, 1, 2, 5, 8, and 10 mM of magnesium chloride following the protocol described above. Similarly, to determine the impact of pH on GUS activity, pNPG was incubated with fecal enzymes in KPI buffer without Mg^2+^ at different pH to determine the relative reaction rates. The results were read at 405 nm after reaction and the relative reaction rates were calculated using the OD values.

In order to determine wogonoside hydrolysis using buffers with different concentrations of Mg^2+^ or at different pH, wogonoside at three different concentrations was incubated with fecal enzymes for 30 min at 37 degrees. Samples were then prepared according to the procedure described above and injected into UPLC to quantitate the metabolite concentrations.

### 2.8. Kinetics of Wogonoside Hydrolysis

In order to determine the kinetics of wogonoside hydrolysis by fecal enzymes, the optimized conditions (without Mg ions, S9 from fresh feces, fecal S9 prepared with sonication, and pH 6.5 for rat and mouse fecal S9 and 7.4 for human fecal S9) were used to obtain reaction rates, which were expressed as amounts of metabolite (i.e., wogonin) formed per min per mg protein (μmol/min/mg). Kinetic parameters were then obtained according to the profile of Eadie–Hofstee plots using the standard Michaelis–Menten equation described as follows:V = (V_max_S)/(K_m_ + S)
where K_m_ is the Michaelis–Menten constant and Vmax is the maximum rate of forming wogonin.

The kinetics parameters were also calculated using Lineweaver–Burk coordinates described in the following.
$$\frac{1}{V}=\frac{k_{m}}{V_{max}}\times \frac{1}{\left[S\right]}+\frac{1}{V_{max}}$$

GraphPad Prism software (version 7.3 for Windows; GraphPad Software, La Jolla, CA, USA) was used. Visual inspection of fitted functions was used to select the best-fit enzyme kinetic model. The results of triplicate incubations are presented as the mean ± S.D.

### 2.9. Statistical Analysis

Two-tailed *t*-tests or one-way ANOVA were used to evaluate statistical differences. Differences were considered significant when *p* values were less than 0.05. Statistical comparisons were performed using GraphPad Prism software (version 7.3 for Windows).

## 3. Results

### 3.1. Confirmation of Wogonoside Metabolite Using UPLC

The metabolite of wogonoside was confirmed by UPLC. The results showed that after incubation, an additional peak was observed at the retention time of 5.08 min in UPLC analysis (Figure 1A). The UV spectrum of this additional peak (Figure 1C) was similar to that of wogonoside (Figure 1B), suggesting that the skeleton of the metabolite is similar to that of wogonoside. Additionally, the retention time of the metabolite peak is identical to that of the standard compound wogonin. When wogonin was spiked into the sample, the peak at RT = 5.08 min increased accordingly. Therefore, this metabolite is identified as wogonin.

Quantification was conducted in UPLC. The test linear response range for wogonoside and wogonin was 0.09–100 μM. Analytical methods for each compound were validated for inter-day and intra-day variation using six samples at three concentrations (40, 15, and 0.5 μM). Precision and accuracy for both compounds were in the acceptable range of 0.11–3.97% and at 88.19–104.62%, respectively.

### 3.2. Total Protein Concentrations Were Different between Sonication and Suspension Preparations

The protein concentration was quantified using the BCA kit following the protocols provided by the manufacturer. The results showed that the total protein concentrations in the S9 fractions are slightly higher than those suspensions for mice, rats, and humans (Figure 2). Since the same amounts of feces and the same volume of buffers were used in these two procedures, it can be concluded that the protein extraction yield is higher when sonication was applied.

### 3.3. Preparation Method Affected GUS Activity

In order to determine the impact of fecal enzyme preparation on GUS activity, pNPG and wogonoside were incubated with enzymes prepared from different species with or without sonication. The results showed that the relative reaction rates of pNPG were significantly higher using enzymes prepared with sonication in all three species. Similarly, the wogonoside hydrolysis rates were also higher using enzymes prepared with sonication in all three species at all three concentrations (Figure 3).

### 3.4. Feces Collection Time Affected Enzyme Activity

In order to determine the impact of feces collection time on GUS activity, pNPG and wogonoside were incubated with enzymes prepared from rat feces collected at different times. The results showed that pNPG relative hydrolysis rates were highest when fresh feces were used. The activity was decreased gradually on day 1 and 7 (Figure 4). Similar results were observed in wogonoside hydrolysis. The hydrolysis rates at all three tested wogonoside concentrations were significantly higher when using enzyme prepared from fresh feces than those of day 1 and 7. Additionally, enzyme prepared using day 1 feces was significantly higher than that of day 7 (Figure 4). These findings suggested that the total activity is decreased once the feces is out of the colon and fresh feces should be collected to avoid activity loss probably due to the death of bacteria during storage in the cages. Additionally, activity loss against wogonoside (up to 47.6%) is greater than that of pNPG (<10%), suggesting that GUS composition could be changing over time.

### 3.5. Magnesium Ions Affected Enzyme Activity

Since magnesium ion significantly affects UGT activity, which is the reverse reaction of glucuronide hydrolysis [29], we determined the impact of magnesium ions on GUS activity by incubating the substrates with buffers containing different concentrations of magnesium ions. The results showed that the relative hydrolysis rates for pNPG were slightly different using buffers with different concentrations of Mg^2+^ (<50%, Figure 5). We then determined the impact of magnesium on wogonoside hydrolysis and the results showed that the hydrolysis rates were also slightly different at different concentrations (<30%, Figure 5). These findings suggest that the impact of magnesium ions on GUS activity was minor, which is different from its reverse glucuronidation reaction mediated by UGTs with 5-fold difference at different concentrations of Mg^2+^ [29].

### 3.6. Buffer pH Value Affected Enzyme Activity

In order to determine the impact of buffer pH on GUS activity, pNPG and wogonoside were incubated with different enzymes in KPI buffers at different pH. The results showed that, for rats and mice, the relative hydrolysis rates of pNPG were highest at pH 6.5 and for humans the highest rate was at pH 7.4 (Figure 6). For wogonoside, the results are as similar as those of pNPG: the hydrolysis rates were highest at pH 6.5, then 5.5 at three different concentrations for both mouse and rat, while pH 7.4 is the best pH to facilitate wogonoside hydrolysis for humans among these tested pH (Figure 6). The favorite pH for these fecal enzymes was calculated using functional dependency calculation, assuming that the impact of pH on enzyme activity follows normal distribution. The results showed that the average favorite pH at different substrate concentrations were 6.08, 5.96, and 6.41 for mouse, rat, and human fecal enzymes, respectively (Figure 6E).

### 3.7. Hydrolysis Rates Are Different across Species

In order to compare the GUS activity across species, pNPG at 1 mM (final concentration), which is a typical concentration used in literature, was incubated with enzymes prepared from mouse, rat, and human feces. The results showed that the relative hydrolysis rates were different across these three species. Rat and human enzymes are 138% and 175%, respectively, of that of the mouse.

In order to quantitatively further describe GUS activity, wogonosides at different concentrations were incubated with the three types of enzymes. The metabolic rates and the kinetic parameters were calculated (Figure 7). The results showed that the hydrolysis followed classic Michaelis–Menten kinetics for these three species. The Vmax for mouse, rat, and human fecal S9 fractions was 2.37 ± 0.06, 4.42 ± 0.12, and 5.17 ± 0.16 μmol/min/mg and the Km were 6.51 ± 0.71, 6.51 ± 0.71, and 0.34 ± 0.047 μM, respectively. The kinetic parameters were also calculated using the Lineweaver–Burk equation and the results showed that the Vmax and Km values were slightly different with those calculated by Michaelis–Menten equation.

### 3.8. Hydrolysis Rates Are Highly Different Across Individuals

We also prepared fecal enzymes using feces collected from different rats and incubated the fecal enzymes with wogonoside at three different concentrations to determine individual variation. The results showed that the hydrolysis rates are highly different across individuals. The reaction rates in rat #6 were the lowest and in rat #9 were the highest. The maximal difference between these two rats were 6.15-fold, 5.38-fold, and 4.51-fold at low (5 μM), medium (15 μM), and high (40 μM) concentrations, respectively (Figure 8).

## 4. Discussion

Different protocols have been used to evaluate microbial GUS activity and the lack of standardization of condition is one of the main factors affecting interstudy variability. In this study, we determined the impact of multiple factors on microbial GUS activity. The standard GUS probe pNPG, which is believed to be hydrolysable by most of the GUS isoforms, was used as a substrate to evaluate the GUS activity globally. Additionally, wogonoside, which is a naturally occurring flavonoid glucuronide that can be hydrolyzed by certain types of GUS, was used as the substrate to evaluate the potential composition changes in the fecal S9. The results showed that sonication during enzyme preparation increased the yield of enzyme extraction from feces (Figure 2), resulting in enhanced GUS activity for pNPG and wogonoside (Figure 3). Additionally, buffer conditions (i.e., pH values and Mg^2+^ concentrations) (Figure 5 and Figure 6) and time of feces collection (Figure 4) also affected GUS activity. We also found that glucuronides hydrolysis rates were different across mice, rats, and humans (Figure 7) and glucuronides hydrolysis rates were highly different across individual animals (Figure 8), even if they were housed at the standard condition and ate the same diet.

In general, drug metabolism by intestinal microflora is determined by incubating drugs with fecal enzymes, even though there is no assurance that the results obtained in in vitro studies could fully reflect the metabolism in vivo [30,31]. In this study, we prepared fecal enzymes using suspension and sonication, which are two methods commonly used in literature, to determine microbial GUS activity [18,19,26,32,33]. The results showed that with the sonication method, the yields of the total protein are higher than with the suspension method for mice, rats, and humans feces (Figure 2). In order to determine the potential impact of sonication on protein activity, we incubated pNPG and wogonoside with the fecal enzymes prepared using the above two methods. The results showed that, for both pNPG and wogonoside, the hydrolysis rates are significantly higher using enzymes prepared with sonication (Figure 3). These findings suggested that sonication could increase GUS extraction yield without affecting enzyme activity in fecal enzyme preparation. We also found that GUS activity was decreased gradually when feces were exposed to air at room temperature (Figure 4), supporting the assumption that fresh feces should be used for fecal enzyme preparation. Additionally, activity loss against wogonoside (up to 47.6%) is greater than that of pNPG (<10%), suggesting that GUS composition could be changing over time.

It was reported that magnesium is by far the most frequently found metal ion cofactor in enzymatic systems and is associated with enzyme activity in mammalian cells [29] and this is probably because magnesium ions can form stable complexes with phosphate-containing species, including ATP [34]. We determined the microbial GUS activity using KPI buffers containing different concentrations of magnesium ions. The results showed that magnesium ion concentration slightly (<33%) affected GUS activity for both pNPG and wogonoside hydrolysis (Figure 5). This result suggested that mammalian enzyme reaction might differ from those in bacteria. Other than magnesium, buffer pH is also an important factor affecting enzyme activity [35]. The impact of pH on microbial GUS activity has never been reported. We then determined the impact of buffer pH on pNPG and wogonoside hydrolysis. The results showed that GUS activities for both pNPG and wogonoside were highly different at different pH (Figure 6), which is similar to the UGT activity [35]. In addition, different species have different favorite pH values. The rat and mouse fecal GUS were most active at pH 6.5 (calculated, 6.08 and 5.96 for mouse and rat, respectively), while human fecal GUS was most active at pH 7.4 (calculated 6.41) and this is likely due to the pH value in the gut of the host being different, since it was reported that human colon pH is around 6.0–7.4 and as for the mouse and rat the values are 5.0–6.5 [36,37].

In preclinical studies, individual variation is another concern. We determined the variation across individuals using rat feces collected from 10 inbred rats housed under the same condition and fed with the same diet. The results showed that fecal GUS difference across individuals is significant (4~6 folds). Although microbial GUS activity is highly different across individual rats, this variation did not exceed the individual variations of phase I and II enzymes (e.g., CYPs and UGTs) activities [38,39]. In order to determine precision activity, researchers should pay more attention towards individual variation as it has never been reported in literature.

The difference across species is frequently reported in drug metabolism studies and the translational potential is always a challenge in preclinical models [40,41]. In order to further elucidate the difference, we determined wogonoside hydrolysis kinetics using rats, mice, and humans fecal S9 fraction under the optimized conditions. The results showed that human fecal enzyme was more active than those of rat and mouse (Figure 7). The Km of human fecal S9 is 10–20-fold less than that of rat or mouse, which is not supervised as the human intestinal microbiome is different from that of rodents [42]. Interestingly, the order of fecal GUS activity in these three species is similar to fecal reduction activity reported previously using a bacteria culture system [43]. For wogonoside, the difference in Vmax is only 2–3-fold, which is less than individual variations (Figure 8). The kinetic parameters were also calculated using the Lineweaver–Burk plot and the results showed that the parameters were slightly different between these two calculations (Figure 7). More studies in the future can be conducted to further elucidate whether there exists any substrate/metabolites inhibition or whether there were any inaccurate measurements using the conditions optimized in this study. For the GUS activity probe pNPG, the difference is only 50%. These findings suggested that the composition of GUS isoform in feces could be highly different.

## 5. Conclusions

The results showed that experimental conditions highly affect the intestinal microbial GUS activity within in vitro studies. The current data showed that sonication should be used when preparing fecal enzymes and fresh feces should be used to ensure optimal activities and accurate evaluations. Additionally, the Mg^2+^ is unnecessary for the buffer system, but pH should be carefully selected based on the species. Intra-individual and species variations in fecal GUS activity are significant and pool fecal samples are encouraged in future studies unless individual variation is taken into consideration.

## Figures and Tables

**Figure 1 pharmaceutics-13-01043-f001:**
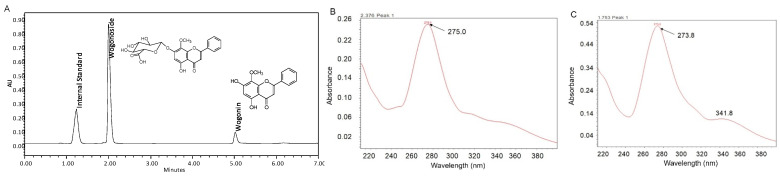
Chemical structures, UV, and a representative UPLC chromatogram of wogonoside and wogonin. Analysis was performed using a Waters Acquity UPLC with photodiode array detector (**A**) a representative chromatogram of wogonoside, wogonin, and I.S.; (**B**) UV spectrum of wogonoside; (**C**) UV spectrum of wogonin.

**Figure 2 pharmaceutics-13-01043-f002:**
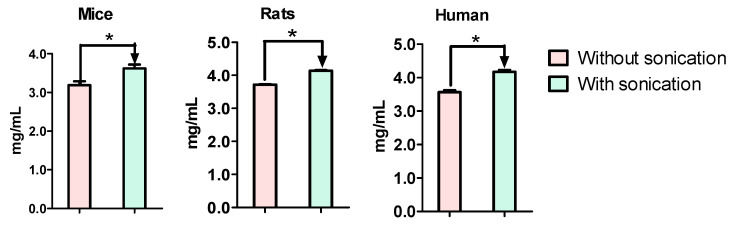
Total protein concentrations were prepared from feces with or without sonication. Feces were homogenized in 50 mM KPI buffer (pH 7.4) with vigorous vortex or sonication (550 w sonicator), followed by centrifugation at 9000× *g* at 4 °C for 15 min (n = 3, * *p* < 0.05, *t*-test).

**Figure 3 pharmaceutics-13-01043-f003:**
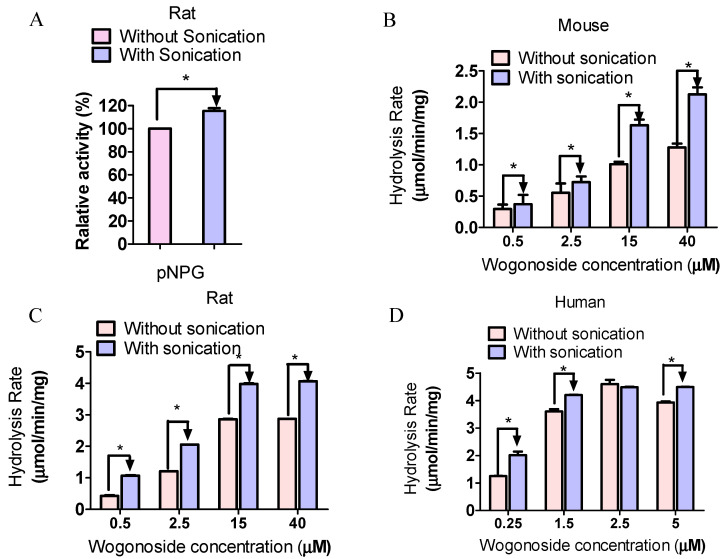
The impact of enzyme preparation on pNPG (**A**) and wogonoside hydrolysis (**B**, mouse; **C**, rat; **D**, human). pNPG (final concentration, 1 mM) and/or wogonoside were incubated with the enzymes for 30 min at 37 °C (n = 3, * *p* < 0.05, *t*-test).

**Figure 4 pharmaceutics-13-01043-f004:**
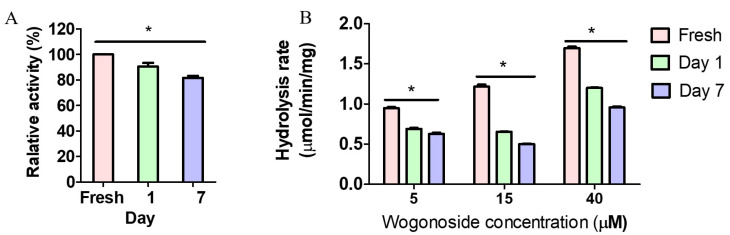
Impact of feces collection time on pNPG (**A**) and wogonoside (**B**) hydrolysis using rat fecal enzyme. The pNPG (final concentration, 1 mM) and/or wogonoside were incubated with the enzymes for 30 min at 37 °C (n = 3, * *p* < 0.05, One-Way ANOVA). The results showed that activity of fresh feces is better.

**Figure 5 pharmaceutics-13-01043-f005:**
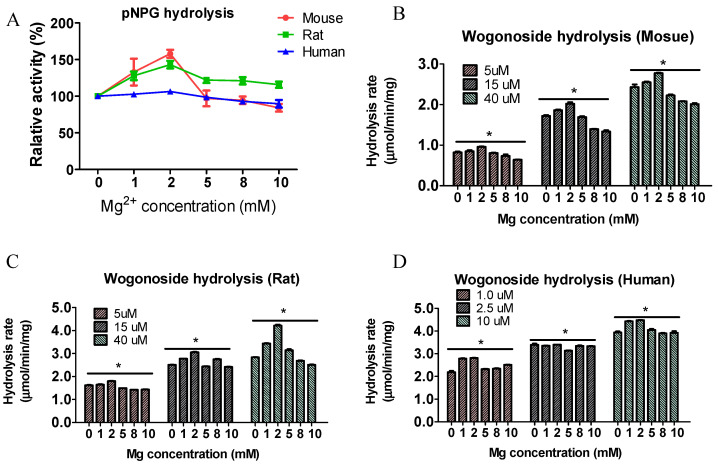
Impact of Mg^2+^ on pNPG (**A**) and wogonoside hydrolysis (**B**, mouse, **C**, Rat, **D**, Human). The pNPG (final concentration, 1 mM) or wogonoside was incubated with the enzymes in the buffers containing different concentrations of Mg^2+^ for 30 min at 37 °C (n = 3, * *p* < 0.05, One-Way ANOVA). The results showed that Mg^2+^ is not a critical co-factor for GUS enzyme.

**Figure 6 pharmaceutics-13-01043-f006:**
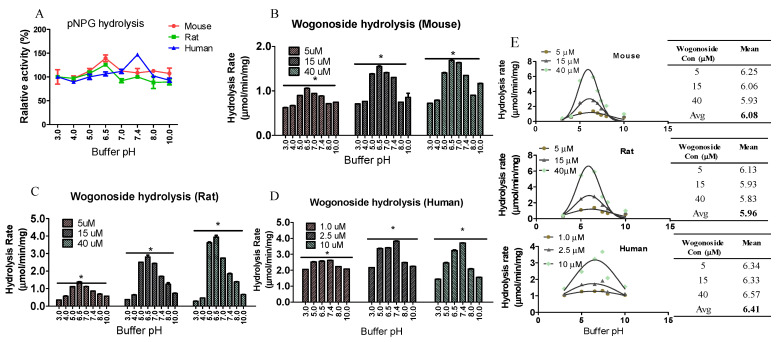
Impact of buffer pH on pNPG (**A**) and wogonoside (**B**, mouse, **C**, Rat, **D**, Human) hydrolysis. The pNPG (final concentration, 1 mM) or wogonoside was incubated with the enzymes in buffers at different pH for 30 min at 37 °C (n = 3, * *p* < 0.05, One-Way ANOVA). The results showed that pH 7.4 and 6.5 are the favorite pH for human and rodent fecal S9, respectively. The favorite pH for these fecal enzymes was calculated using functional dependency calculation; (**E**) Avg, average, Gaussian distribution.

**Figure 7 pharmaceutics-13-01043-f007:**
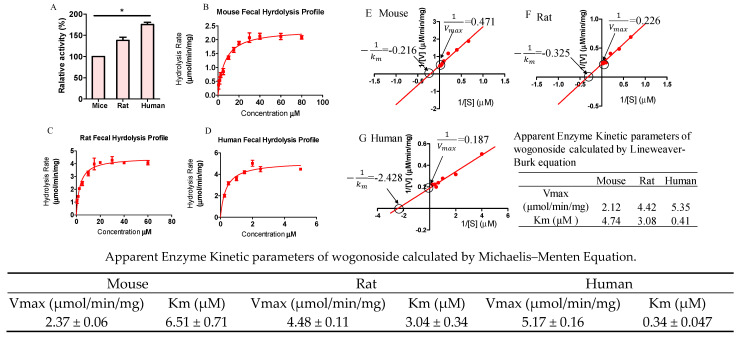
pNPG (**A**, * *p* < 0.05, One-Way ANOVA) and wogonoside (**B**, mouse; **C**, rat; **D**, Human) hydrolysis using enzymes from different species. (**B**–**D**) were Michaelis–Menten plots and (**E**–**G**) were Lineweaver–Burk plots. Wogonosides were incubated with the enzymes in buffers for 30 min at 37 °C (n = 3). The results showed that human fecal S9 is more active than that of rodents.

**Figure 8 pharmaceutics-13-01043-f008:**
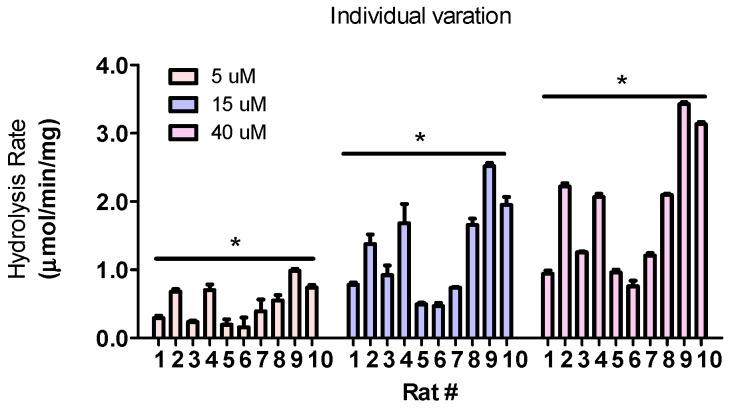
Wogonoside hydrolysis individual variation. Wogonoside were incubated with enzymes prepared from individual rats in KPI buffer (50 mM, pH 6.5) for 30 min at 37 °C (n = 3, * *p* < 0.05, One-Way ANOVA).

## Data Availability

The data presented in this study are available in the article.

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
