# Peer review of "Glucuronides Hydrolysis by Intestinal Microbial β-Glucuronidases (GUS) Is Affected by Sampling, Enzyme Preparation, Buffer pH, and Species"

_pharmaceutics, 2021, doi:10.3390/pharmaceutics13071043_

Round 1

Reviewer 1 Report

The peer-reviewed article presents a study on hydrolysis of glucuronide substrates by intestinal microbial β-Glucuronidases. This topic is of great interest to a wide range of readers due to growing interest in the intestinal microflora in recent years. Another important aspect investigated in this work is the standardization of kinetic data related to enzymatic systems from raw biomaterials. This aspect is important for quantitative studies of various biosystems and natural materials, but the problem has not yet been resolved. In the article, the authors presented a huge amount of experimental data on the hydrolysis of 4-nitrophenyl β-D-glucopyranoside and natural glucuronide wogonoside under various conditions, including the time of collection of feces, buffer conditions, and individual and species variations.

I think that the article can be improved taking into account the following comments and suggestions:

- It is better to bring the Michaelis-Menten equation in the classical form (the multiplication sign is usually not written): V=VmaxS/(Km+S)

- The designations on the axes in Fig. 1B,C are not visible.

- The kinetic parameters Vmax and Km were determined directly from the Michaelis-Menten equation. The representation of data in reciprocal coordinates (Lineweaver-Burk coorditates) was not used. I wonder what would be the error in determining Vmax and Km if reciprocal coordinates were used. This is especially important for the humans (Fig. 7D), where there are not many experimental points. Please discuss this point.

- The pH profiles for hydrolysis of wogonoside were provided by the authors (Fig 6). The question arises as to how best to determine the pH corresponding to the maximum values of these functions? If we use a functional dependence, for example, a bell-shaped one, then the pH of the maximum can be calculated with greater accuracy using all the obtained experimental points. Please discuss why this opportunity was not used.

Author Response

We thank this reviewer’s effort and appreciate this reviewer’s acknowledgment of our innovative work. We have already revised the manuscript accordingly. Major changes are in blue in the version. Please see the attached file for our response. 

Reviewer 2 Report

The term "Experimental Conditions" used in the title; it would be better to replace it more specifically.

The abbreviation "GUS" appears repeatedly without rules, here and ther with its full name, it should be fixed. Actually it is vaguely revealed which word GUS is abbreviated for, which makes it very difficult to read. 

Major factors authors concerned about looks like "sampling, pH and species variation", if the variables are important to take, it is better to study among the variables as well. 

Author Response

We thank this reviewer’s effort and appreciate this reviewer’s tolerance for our grammatical errors. We have already revised the manuscript accordingly. Major changes are in blue in the version. Please see the attached file for our response. 
